# Hair Growth-Promoting Activities of Glycosaminoglycans Extracted from the Tunics of Ascidian (*Halocynthia roretzi*)

**DOI:** 10.3390/polym14061096

**Published:** 2022-03-09

**Authors:** Therese Ariane N. Neri, Grace N. Palmos, Shin Young Park, Tae Sung Jung, Byeong-Dae Choi

**Affiliations:** 1Nutrition Chemistry Laboratory, Department of Seafood Science and Technology, The Institute of Marine Industry, Gyeongsang National University, Tongyeong 53064, Korea; yanie@gnu.ac.kr; 2Institute of Fish Processing Technology, College of Fisheries and Ocean Sciences, University of the Philippines Visayas, Iloilo 5023, Philippines; gnpalmos1@up.edu.ph; 3Food Hygiene Laboratory, Department of Seafood Science and Technology, The Institute of Marine Industry, Gyeongsang National University, Tongyeong 53064, Korea; sypark@gnu.ac.kr; 4Laboratory of Aquatic Animal Diseases, Research Institute of Natural Science, College of Veterinary Medicine, Gyeongsang National University, Jinju-daero, Jinju 52828, Korea; jungts@gnu.ac.kr

**Keywords:** ascidian tunic, glycosaminoglycans, hair follicle, C57BL/6 mice, hair growth

## Abstract

Throughout the ages, hair has had psychological and sociological importance in framing the personality and general appearance of an individual. Despite efforts to solve this problem, no groundbreaking measures have been proposed. Glycosaminoglycans (GAGs) and associated proteoglycans have important functions in homeostatic maintenance and regenerative processes of the skin. However, little is known about the role of these molecules in the regulation of the hair follicle cycle. Three fractions (F1, F2 and F3) were obtained after separation and purification of GAGs from ascidian tunics. F1 was observed to contain a small amount of amino sugar while high contents of galactose and N-acetylglucosamine were noted in F2 and F3. 2-acetamido-2-deoxy-3-O-(β-D-gluco-4-enepyranosyluronic acid)-6-O-sulfo-D-galactose (∆Di-6S) and 2-acetamido-2-deoxy-3-O-(β-D-gluco-4-enepyranosyluronic acid)-4-O-sulfo-D-galactose (∆Di-4S) were the main disaccharide components. F3 exhibited the highest proliferation activity on human follicle dermal papilla (HFDP) cells. In addition, mixed samples (FFM) of F2 and F3 at different concentrations showed peak activities for five days. After cell culture at a concentration of 10 mg/mL and dihydrotestosterone (DHT), the inhibition effect was higher than that for Minoxidil. Application of 10 mg of FFM to the hair of mice for 28 days resulted in a hair growth effect similar to that of Minoxidil, a positive control.

## 1. Introduction

Hair is a specialized derivative structure of skin and a defining characteristic of the human integumentary system. The hair root is buried inside the epidermis and enclosed within a hair follicle (HF), consisting of epithelial components (matrix and outer-root sheath) and dermal components (dermal papilla and connective tissue sheath) [1]. Growth of human HF and formation of hair shaft requires terminal differentiation-associated cell cycle arrest of highly proliferative matrix keratinocytes [2]. Human hair matrix keratinocytes follow a path of active cell cycling, arrest and terminal differentiation that showcase the human HF as an excellent, clinically relevant model system for research on cell cycle physiology of human epithelial cells within their natural tissue habitat [3].

Hair loss is a common hair disorder, especially in males. Oxidative stress induced by the environment, aging and stress causes the onset and progression of hair loss [4,5]. Despite the development of many drugs and shampoos for prevention and treatment of hair loss, no breakthrough has yet been made. Because side effects have been discovered for the recently certified products, Finasteride and Minoxidil, there is a pressing need for developing new products to replace them [6,7]. Hair loss can occur throughout the scalp and the entire body. Many growth factors, cytokines and hormones have been associated in the regulation of the hair growth cycle, but the specific mechanism of hair loss has not been clarified. The known factors related to hair loss, so far, include genetic background, hormonal changes, pathological conditions, aging and side effects of a drug [8]; wherein, heredity and age are the leading causes. Although hair loss is not a life-threatening condition, it can cause significant impairment to the quality of life. The drugs currently approved by the FDA for treatment of androgenetic alopecia are oral Finasteride (Propecia), a 5-α reductase type II inhibitor and Minoxidil, an external agent for hair growth which increases blood flow [9,10].

Glycosaminoglycans (GAGs) are long unbranched polysaccharides containing a repeating disaccharide unit that contain either of two amino sugars N-acetylglucosamine (GlcNAc) or N-acetylgalactosamine (GalNAc) and a uronic acid such as glucuronate or iduronate. Different disaccharide units combine to form the different kinds of GAGs, many of which are mono- or di-sulfated at different sites and exhibit specific biological activities depending on the structure [11]. Ascidians, commonly known as sea squirts or tunicates, are one of the primary seafood commodities in Korea. Sourced from both wild and cultured environments, and locally known as “*mongge*”, production volume of sea squirts is averaging at approximately 10,000 tons from 2006 to 2013 and increased to 31,326 tons in 2020 [12]. Ascidians are mostly sold as fresh produce in the local market, with the internal organs removed or sometimes sold together with the meat. The edible body is usually separated from the thick and tough tunic before being processed as a frozen good [13]. The inedible tunics become waste products in the production line and are mostly just discarded back to the sea or left to dry under the sun. Therefore, finding uses for them is important in order to reduce environmental impact and for added income. Identifying novel components with potential functional and medical applications is a method for utilizing the tunics [14]. In this study, using C57BL/6 mice and cell viability testing, we provide evidence of the hair growth-facilitating activity of fractionated FFM from ascidian tunics.

## 2. Materials and Methods

### 2.1. Materials and Chemicals

Ascidian (*Halocynthia roretzi*) tunics were purchased from a fresh market in Tongyeong, Korea. The tunics were washed with tap water and kept at −40 °C. Chondroitin AC lyase (EC 4.2.2.5) from *Arthrobacter aurenses*, chondroitin ABC lyase (EC 4.2.2.4) from Proteus vulgaris, monosaccharide standards, 1,9-dimethylmethylene blue, glucuronolactone and toluidine blue-O were obtained from Sigma-Aldrich (St. Louis, MO, USA); DEAE-Sepharose was obtained from GE Healthcare Bio-Sciences AB (Uppsala, Sweden). Human follicle dermal papilla (HFDP) cells (C-12071) were purchased from PromoCell GmbH (Heidelberg, Germany). Cells were cultured in follicle dermal papilla cell growth medium (C-26501, PromoCell GmbH, Heidelberg, Germany) supplemented with 10% fetal bovine serum (FBS, Thermo Fisher Scientific, Waltham, MA, USA), 50 units/mL of penicillin and 50 mg/mL of streptomycin. All the other reagents were of analytical grade.

### 2.2. Extraction and Proximate Analysis of Ascidian Tunic Mucopolysaccharides

One kilogram of tunics was pressure extracted at 121 °C for 3 h with 10 volumes of distilled water and concentrated in a vacuum evaporator until a Brix value of 5–6 was obtained. The concentrates were centrifuged for 20 min at 8000 rpm, and the supernatants mixed with three volumes of 95% ethanol three times at room temperature. The filtrates (Whatman No. 2) were lyophilized and stored at −40 °C until further analyses. Moisture content was analyzed using oven constant drying at 105 °C. Crude protein was quantified using the micro-Kjeldahl method and crude lipid was determined using the Bligh and Dyer method [15]. Ash content was evaluated after heating in a muffle furnace at 550 °C. Phenol-sulfuric acid assay was used for the determination of total carbohydrate concentration [16]. The sulfation degree of polysaccharides was estimated via turbidimetric method (BaCl^2^/gelatin) [17].

### 2.3. Separation of Glycosaminoglycans Fractions Using Ion-Exchange Chromatography

The GAGs extracted from the ascidian tunics (~250 mg) were applied to a DEAE-Sepharose Fast Flow column (5 × 30 cm, GE Healthcare Bio-Sciences AB, Uppsala, Sweden) equilibrated with 0.5 M sodium acetate buffer (pH 6.0). The column was developed using a linear gradient of 0–1.5 M NaCl in the same buffer. The flow rate of the column was 5 mL/min and fractions were collected and assayed by metachromasia using 1,9-dimethylmethylene blue for sulfated GAGs [18] and by carbazole reaction for hexuronic acid [19]. The positive fractions were pooled, dialyzed against distilled water and lyophilized. Each fraction was reapplied to a newly packed DEAE-Sepharose Fast Flow column (5 × 30 cm) and re-purified as described above.

### 2.4. Analysis of Neutral and Amino Sugar Contents by High-Performance Anion Exchange Chromatography-Pulsed Amperometric Detector (HPAEC-PAD)

Neutral and amino sugar contents of the eluted fractions (F1, F2, F3) from the ion-exchange chromatography were analyzed using HPAEC-PAD System (Dionex Co., Bannockburn, IL, USA). Sample solutions (10 µL) of the fractions were passed through CarboPac PA-1 column (4.5 × 250 mm, Dionex Co., Bannockburn, IL, USA) and eluted with isocratic gradient of 18 mM NaOH at a flow rate of 1 mL/min, for neutral sugars. HPAEC-PAD experiments were performed in triplicate for each sample. Reference standard solutions were analyzed in duplicate for the standard curve. Good standard curve linearity was observed (r ≥ 0.990) [20].

### 2.5. HPLC Analysis for Disaccharide Composition

HPLC system from Shimadzu equipped with a Spherisorb SAX column (5 μm, 4.6 × 250 mm, Waters) fitted with a guard cartridge (Waters Spherisorb, 5 μm, 4.6 × 10 mm) was used for analysis of disaccharides after digestion of the GAGs fractions with chondroitin ABC and/or AC lyase (Sigma-Aldrich Co., St. Louis, MO, USA). Elution was performed at 1.5 mL/min in isocratic mode from 0 to 5 min with 50 mM NaCl (pH 4.0) and then in linear gradient from 5 to 25 min, starting with 50 mM NaCl (pH 4.0) and ending with 76% 50 mM NaCl (pH 4.0) and 24% 1.2 M NaCl (pH 4.0). Disaccharide peaks were detected at 232 nm and quantified by external calibration [21].

### 2.6. HFDP Cell Culture and Viability Test

HFDP cells were cultured in follicle dermal papilla cell growth medium (PromoCell GmbH, Heidelberg, Germany) containing 10% FBS, 50 units/mL of penicillin and 50 mg/mL of streptomycin. HFDP cells were distributed at a concentration of 1 × 10^4^ cells/well to a 96-well plate and cultivated in a CO^2^ incubator adjusted to 37 °C with 95% humidity and 5% CO^2^. For cell viability, cells were treated with F1, F2, F3 (1, 10, 50 and 100 μg/mL) and hyaluronic acid (1 μg/mL) for 24 h. MTS solution (0.5 mg/mL, CellTiter 96^®^ Aqueous One Solution Cell Proliferation Assay kit) was added to each well for 4 h (5% CO^2^ at 37 °C), and 100 μL of DMSO was added to solubilize purple formazan crystals. The mixed samples (F2 + F3 fraction, FFM) were dissolved in the new growth medium so that the final concentration was 1, 10, 50 and 100 μg/mL, followed by treatment with cytoplasm, and cultivation for five days. The proliferation rate of cells was measured using the Cell Titer 96^®^ Aqueous One Solution Cell Proliferation Assay purchased from Promega Co. (Madison, WI, USA). The absorbance was measured at 490 nm using the Spectramax M2 Microplate Reader (Molecular Devices, Sunnyvale, CA, USA) [22].

### 2.7. Inhibition of Dihydrotestosterone (DHT) Production

Liver tissue of Sprague Dawley rats (*n* = 3, 6 g each) was homogenized with 12 mL of PBS; in total, 1 mL of this homogenate was applied with the fractions and allowed to react at 37 °C for 15 min. Then, 1 mL of buffer (50 mM Na_2_HPO_4_, 0.25 M sucrose) was added, and a vortex was used for mixing. The mixture was centrifuged at 4 °C, and the supernatant was used as sample. An ELISA kit for DHT was used (Wuhan USCN Business Co. Ltd., Wuhan, China) to measure the degree of DHT production; in total, 50 μL of the standard and samples were dispensed and incubated on 96-well plates included in the kit at 37 °C for 1 h. After incubation, the solution was removed and 50 μL of detection reagent A was added, followed by incubation at 37 °C for 1 h. To this, 100 μL of detection reagent B was dispensed, and the mixture was incubated at 37 °C for 30 min. After the reaction was completed, the supernatant was removed, followed by washing five times with buffer. After washing, 90 μL of the substrate solution was added; the plate was wrapped with foil, and incubated at 37 °C for 15 min. To end the reaction, 50 μL of stop solution was added, and the absorbance was measured at 450 nm.

### 2.8. Hair Regrowth Activity in C57BL/6 Mice

Seven-week-old male C57BL/6 mice were purchased from OrientBio Co. (Seoul, Korea). After acclimatization to the breeding environment for seven days, the animals were divided into four randomized groups (*n* = 4); normal saline-, 5 mg of Minoxidil-, and low (5 mg) and high dosage (10 mg) of FFM-treated groups, to investigate the hair growth-promoting activity of fractions via topical application. After shaving the dorsal hair of 8-week-old male C57BL/6 mice, corresponding treatments were applied daily to each group for 4 weeks and hair regeneration was monitored according to Matsuda et al. with slight modifications [23]. Hair regrowth at 7, 14, 21 and 28 days after the start of topical application was recorded by photography using a digital camera once per week for four weeks, until a significant difference was observed from each group.

### 2.9. Statistical Analysis

The results were presented as means ± standard deviation of three replicates. Statistical evaluations were performed using Duncan’s multiple range test or one-way analysis of variance (ANOVA) with the Tukey test (MINITAB, Release 14.20). All *p*-values less than 0.05 were considered statistically significant (*p* < 0.05 is represented by *, *p* < 0.01 is represented by **).

## 3. Results and Discussion

### 3.1. Physicochemical Properties of the Ascidian Tunic

Polysaccharides found in marine organisms are known to have effects on inflammatory cytokines and chemokines, angiogenesis-related growth factors such as fibroblasts growth factors-1 and neovascularization [24,25,26]. In this study, approximately 5.3% of crude GAGs were extracted from ascidian tunics which contained 35.9% protein, 32.0% ash and 27.8% carbohydrates, as shown in Table 1. The carbohydrate content consisted of 57.1% mucopolysaccharides and 3.6% sulfates. Hong et al. [27], performed different enzyme-extractions of GAGs from the ascidian tunic which yielded 2.3–3.1% of crude GAGs extract. Chemical analyses of the GAGs extract in Hong et al.’s study showed that chondroitin sulfate content was approximately 25.7%, which is relatively similar to the results of this experiment. Additionally, the sugar content was composed mainly of galactose at 57.8%, followed by N-acetylgalactosamine (GalNAc) at 15.8% [27]. Meanwhile, the high ash content found in the crude GAGs extract may be attributed to the fact that the ascidians are filter feeder benthic species, with amounts of minerals such as potassium, sodium and calcium providing structure for the tunic [28].

### 3.2. Purification of Ascidian Tunic GAGs Extracts by Ion-Exchange Chromatography

Crude mucopolysaccharide extracts were purified by ion-exchange chromatography using sodium chloride in acetate buffer (pH 6.0). Three active fractions (F1, F2 and F3) were found and monitored for presence of sulfated GAGs and uronic acids using metachromasia and carbazole assays, respectively. Yields of each fraction were: 15.7 mg%, for F1; 29.0 mg%, for F2; and 36.0 mg%, for F3 (Table 2). Uronic acids were present at various concentrations in all three fractions, while only F2 and F3 were found to contain sulfated GAGs and, thus, were used for further analyses. A relatively high content of sulfated GAGs (18.2%) was noted in F3 when compared to the GAGs extracted from other sources such as shark fin (9.60%), crocodile (14.84%), ray (5.27%) and chicken keel (14.08%) [29].

### 3.3. Sugar Compositions of Ascidian Tunic GAGs Fractions

Neutral sugar and amino sugar analysis of fractions showed that the crude extract was mainly composed of galactose with small amounts of glucose and mannose while the major amino sugar was glucosamine (Figure 1, Table 3). F1 only contained glucose, with trace amounts of amino sugar. On the other hand, galactose was the main neutral sugar component of the more active F2 and F3, which contained lesser amounts of glucose and mannose. In addition, the presence of rhamnose and fucose was also observed in F2. N-acetyl glucosamine was the major amino sugar component of both F2 and F3, with higher N-acetyl glucosamine content in F3 than in F2. The purity of each fraction was determined based on these results, with the higher purity (92.4%) for F3 than F2 (89.9%). These results also implied the possibility of different GAGs contained in the crude extracts, with F2 as more diverse than F1 and F3.

Plants and marine-sourced polysaccharides vary in their monosaccharide contents. Flaxseed mucilage was extracted using an aqueous process and monosaccharide composition showed that xylose was the most abundant saccharide (21.7%). Comparable lower levels were observed for galactose (8.1%), arabinose (7.9%) and rhamnose (8.4%). Minor monosaccharides were glucose (1.7%), fucose (3.4%) and galacturonic acid (6.3%). Overall, neutral polysaccharides of flaxseed mucilage were estimated to be 72%, composed of xylose, arabinose, galactose and fucose [30]. Meanwhile, a sulfated heteropolysaccharide, fucoidan, was extracted from *Laminaria japonica* which is an important economic alga species in China. Three polysaccharide fractions (LJFF1, LJFF2 and LJFF3) were successfully isolated using anion-exchange column chromatography. Results showed that fucose was the main sugar form in LJFF2, accounting for 76.56% of the total neutral sugar while galactose was the major sugar form in LJFF1 and LJFF3, amounting to 43.30% and 78.54%, respectively. In addition to fucose and galactose, the other monosaccharides found were mannose, glucose, rhamnose and xylose in LJFF1 and mannose in LJFF2, indicating that the chemical composition may have a significant influence on antioxidant activities. Fucose contained in LJFF1, LJFF2 and LJFF3 was 12.40%, 29.26% and 16.85%, respectively. The highest sulfate content (36.67%) was noted in LJFF3 while LJFF1 contained the lowest sulfate content (23.30%) [31].

### 3.4. Disaccharide Composition of Ascidian GAGs as Determined by HPLC

Chondroitinase ABC catalyzes the eliminative degradation of polysaccharides containing (1→4)-β-D-glucoronosyl or (1→3)-α-L-iduronosyl linkages into disaccharides containing 4-deoxy-β-D-gluc-4-enuronosyl groups. This enzyme acts on chondroitin sulfate A and C, chondroitin sulfate B (dermatan sulfate), and at a lesser degree on hyaluronate. The enzyme chondroitinase AC is an eliminase that degrades chondroitin sulfates A and C but not chondroitin sulfate B [32]. Chondroitin sulfates can be differentiated from dermatan sulfate using a combination of chondroitinase ABC and chondroitinase AC digestion. Chondroitinase ABC reacts on chondroitin sulfate A and B, resulting in ΔDi-4S, and on chondroitin sulfate C, resulting in ΔDi-6S, while chondroitinase AC hydrolyzes only chondroitin sulfate A and C, resulting in ΔDi-4S and ΔDi-6S, respectively [33].

Enzyme-digested fractions (F2 and F3) and disaccharide standards were subjected to HPLC analysis to determine the disaccharides components of the polymer and are shown in Table 4. Results showed that both fractions contain ΔDi-6S and ΔDi-4S. The molar ratio (ΔDi-6S:ΔDi-4S, rounded off) of F2 sample digested with chondroitinase ABC and AC was 47:53 and 49:51, respectively, while the molar fraction for F3 was 49:51 and 43:57, respectively. As mentioned previously, ΔDi-6S is the disaccharide component of chondroitin sulfate C (chondroitin-6-sulfate) while ΔDi-4S is the disaccharide component of chondroitin sulfate A (chondroitin-4-sulfate). Because digestion with chondroitinase ABC and chondroitinase AC yielded an almost equimolar ratio and with almost the same values for F2, we can presume that F2 is composed of an equal amount of chondroitin sulfate A and C. For F3, digestion with chondroitinase AC resulted in a slightly higher molar ratio for ΔDi-4S. Since chondroitinase ABC hydrolyzes almost all types of chondroitin sulfate including dermatan sulfate (chondroitin sulfate B), and chondroitinase AC only acts upon chondroitinase A and chondroitinase C, we can deduce that F3 may contain a very small amount of other chondroitin sulfate isomers, or chondroitin sulfate-like compounds. However, in general, we can assume that the main components are chondroitin sulfate A and chondroitin sulfate C.

Although dermatan sulfate (DS) is composed of repeating disaccharide building blocks made up of alternating 4-linked α-L-iduronic acid (IdoA) and 3-linked β-D-galactosamine (GalNAc) units, sulfation patterns can vary among DS from different tissues, cells, or pathophysiological conditions. Nonetheless, most common DSs from mammals have GalNAc mostly sulfated at the C4 position (∼95%) and slightly at the C6 position (∼15%) while composing IdoA units are only occasionally sulfated at the C2 position (∼5%). The same sulfation pattern can be observed in chondroitin sulfates (CS), with the IdoA replaced by glucuronic acid. In ascidians, also known as tunicates or sea squirts, their CS/DSs exhibit different sulfation patterns which vary according to the ascidian species [34].

### 3.5. Cell Proliferation Activity of Ascidian Tunic GAGs on Human Follicle Dermal Papilla (HFDP)

Hair consists of two distinct structures: (1) hair follicle, found beneath the skin and (2) the shaft, the hard-filamentous part that extends above the skin surface. Each human hair follicle independently undergoes the regeneration cycle. Normally, 90% go through anagen (growth phase), 1–2% catagen (regression phase) and 8–9% telogen (resting phase) [32,33]. The independent behavior of each follicle as it passes through the different stages of growth and termination endows each person with the normal volume of hair.

The effect of each ascidian tunic GAGs fraction on the viability of HFDP cells was monitored during a period of 24 h (Figure 2). HFDP cells were with each of the tunic fractions at 1, 10, 50 and 100 μg/mL and compared to the cell-only negative control and to the cells treated with the positive control, hyaluronic acid. After treatment, all fractions did not show toxicity on HFDP cells and cell viability was significantly higher than the control groups in the cells treated with F3 fraction at 50 and 100 μg/mL concentrations. The results indicated the potential of ascidian tunic GAGs extract to promote cell proliferation. Thus, F2 and F3 were combined (FFM) and further examined for proliferative activity on HFDP cells. After 5 days of culture, notable increases in proliferation were observed in cells treated with 50 and 100 μg/mL FFM (21% and 27%, respectively, as shown in Figure 3). This showed that FFM from ascidian tunics promoted cell proliferation, which could also suggest a potential to promote hair growth.

Improved or enhanced cell proliferation could be attributed to the replenishment of the proteoglycan components of the papilla. Proteoglycan moieties, namely: chondroitin sulfate; heparin sulfate; dermatan sulfate; and keratin sulfate, are variably distributed in the dermal part of the hair follicle [35]. Dermal papilla is also enriched with basement-membrane components, including a chondroitin-6-sulfate-containing proteoglycan [36,37]. GAGs moieties are abundantly present during the anagen stage and decrease in quantity as the hair follicle continues to telogen.

### 3.6. Inhibitory Effects of the FFM Fraction on Hormonal Factor Dihydrotestosterone (DHT)

Stimulatory and inhibitory factors, including hormones and pharmaceutical products, influence the hair cycle. DHT, an androgenic hormone produced through catalysis of testosterone by the enzyme 5α-dihydrotestosterone, is implicated as the responsible promoter of androgenetic alopecia [38]. An increased level of DHT causes the shortening of the hair cycle and progressively miniaturizes scalp follicles, which may be due to the atherosclerotic process blocking the microvasculature that nourishes hair follicles [39]. Minoxidil is a DHT inhibitor approved by USFDA and is used in the treatment of androgenetic alopecia. It acts by decreasing the serum levels of DHT, stopping hair loss and stimulating hair regrowth [40]. DHT concentrations in the control and treated groups were measured using an enzyme-linked immunosorbent assay with Minoxidil used as a positive control, (Figure 4). The propylene glycol control group was noted to have DHT concentration of 6.24 ng/mL of. In the Minoxidil-treated group with dose concentrations of 1 and 10 mg/mL, DHT concentrations were 3.01 and 2.38 ng/mL, respectively. Meanwhile, FFM-treated groups yielded DHT concentrations of 1.63 and 1.02 ng/mL, respectively, for 1 and 10 mg/mL treatment dosage. Combined fractions, FFM, extracted from tunics exhibited two-fold better ability to inhibit DHT production than Minoxidil depending on the concentration, indicating the potential and feasibility of FFM ascidian GAGs extract in the prevention of hair loss.

### 3.7. Hair Growth in C57BL/6

The hair cycle is regulated by complex epithelial–mesenchymal interactions. Several growth factors in the dermal papilla and overlying epithelial cells are known to interact with each other during the progression of the hair growth cycle [41]. In this study, the hair growth-promoting activity of FFM was observed when applied topically once a day for 4 weeks to C57BL/6 mice at two different dosages, 5 and 10 mg/day (FFM1 and FFM2, respectively). Minoxidil (5 mg/day) was similarly applied onto the backs of a separate group of mice as a positive control. As shown in Figure 5, from the 14th day, the color of dorsal skin showed a more rapid change to black in the Minoxidil-treated group, suggesting active transition to the anagen phase. Hair growth on the 28th day was more prominent in the FFM2 group compared with those of the control and Minoxidil group.

## 4. Conclusions

In this study, three purified fractions (F1, F2 and F3) were successfully isolated from the crude GAGs extracts of ascidian, *H. roretzi*, tunics using ion-exchange chromatography. Monosaccharide and disaccharide composition confirmed that both F2 and F3 were composed of a mixture of chondroitin sulfates A and C with minor chondroitin sulfate isomers in F3, while F1 contained only glucose. All three fractions showed no toxicity against HFDP cells and a mixture of F2 and F3 fractions (FFM) exhibited improved proliferation in HFDP cells upon treatment. The ascidian mixed fraction, FFM, displayed a potential to delay hair baldness by effectively inhibiting DHT production compared to the non-treated control group. In addition, topical application in mice for 28 days demonstrated that hair regrowth, as well as skin darkening, was more noticeable in the group treated with FFM compared to the normal control group.

## Figures and Tables

**Figure 1 polymers-14-01096-f001:**
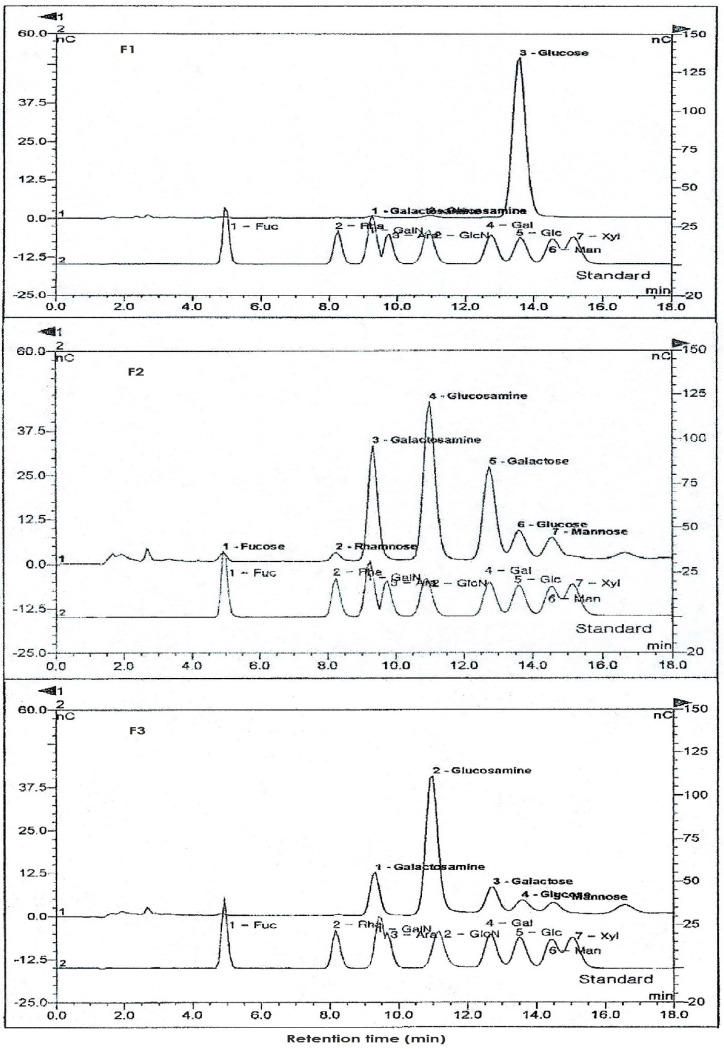
Representative chromatogram of standard monosaccharides and F1, F2 and F3 by HPAEC-PAD. Monosaccharides and glucosamines were separated by RP-HPLC, hydrolyzed in 4 M TFA for 4 h. The amount of each standard monosaccharide and glucosamine was 5 nmol and 1 nmol, respectively (Table 3).

**Figure 2 polymers-14-01096-f002:**
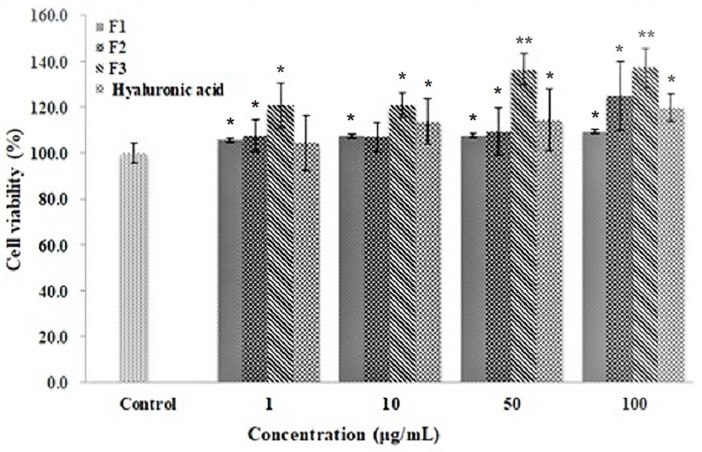
Effect of ascidian tunic GAGs on the viability of human follicle dermal papilla (HFDP) cells. HFDP cells were treated with different concentrations of each fraction (1, 10, 50, 100 μg/mL) and hyaluronic acid (1 μg/mL, HA) for 24 h. Values are mean ± standard deviations of three (*n* = 3). * *p* < 0.05, ** *p* < 0.01 compared with the control.

**Figure 3 polymers-14-01096-f003:**
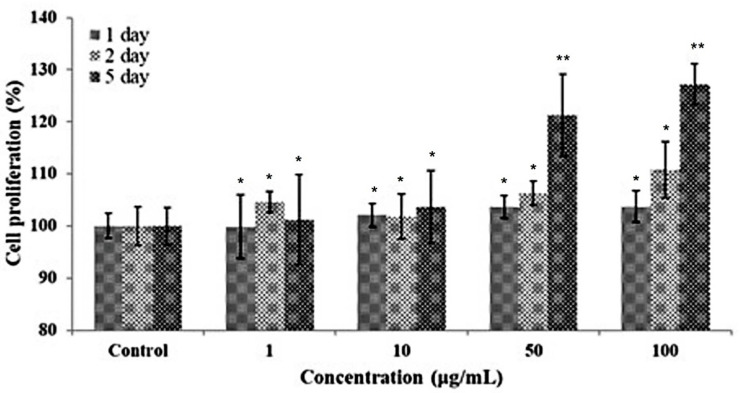
The stimulating effect of FFM (F2 and F3 mixtures) on hair follicle dermal papilla (HFDP) cells. HFDP cells were treated with different concentrations of FFM for five days at different concentrations. Values are mean ± standard deviations of three (*n* = 3) individual experiments. * *p* < 0.05, ** *p* < 0.01 compared with the control.

**Figure 4 polymers-14-01096-f004:**
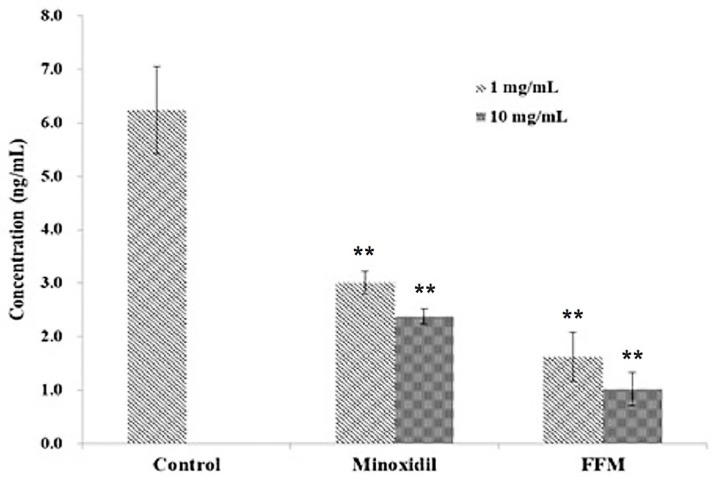
Dihydrotestosterone (DHT) production in rat liver tissue of Minoxidil- and FFM-treated groups at different concentrations. Values are mean ± standard deviations of three (*n* = 3) individual experiments. Control, Propylene glycol; FFM is mixture of F2 and F3 fractions; Minoxidil was used as positive control to evaluate the production of DHT. ** *p* < 0.01 compared with the control.

**Figure 5 polymers-14-01096-f005:**
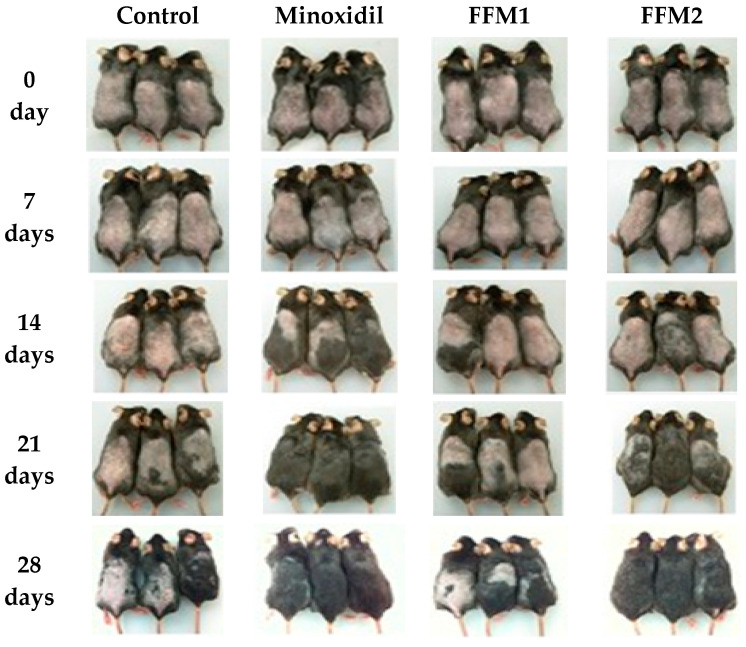
Gross observation of hair regrowth in C57BL/6 mice for 28 days. Each group of mice was treated topically with normal saline or 5 mg of Minoxidil or 5 mg (FFM1) and 10 mg (FFM2) of mixture fraction once a day for consecutive 28 days, respectively. Depilated dorsal skin lesions were photographed after topical application of vehicles. Note: The sample group treated with minoxidil developed an allergic reaction on the 3rd week of treatment, resulting in lesser mice in the said group on the 21st and 28th days of treatment.

**Table 1 polymers-14-01096-t001:** Proximate composition, mucopolysaccharides and sulfates content of extracts from ascidian tunic.

Components	Percentage ^1^ (%)
Yield	5.3 ± 0.3
Moisture	3.1 ± 0.4 ^c^
Crude protein	35.9 ± 2.3 ^a^
Lipid	1.2 ± 0.1 ^c^
Ash	32.0 ± 1.2 ^a^
Carbohydrates	27.8 ± 2.9 ^a,b^
Mucopolysaccharides	57.1 ± 1.5
Sulfate	3.6 ± 0.3

^1^ Values are mean ± standard deviations of three (*n* = 3) measurements with different superscripts in a column in a specific assay vary significantly (*p* < 0.05).

**Table 2 polymers-14-01096-t002:** Yield and purity of the sulfated glycosaminoglycans in mucopolysaccharides from ascidian tunic extracts.

Fractions	Yields (mg%)	Sulfated GAGs (%)
F1	15.7 ± 0.2 ^c^	-
F2	29.0 ± 0.4 ^b^	7.7 ± 0.6 ^b^
F3	36.0 ± 0.7 ^a^	18.2 ± 1.5 ^a^

Values computed based on dry weight of ascidian tunics and presented as mean ± standard deviations of three (*n* = 3) measurements, with different superscripts in a column indicating significant difference (*p* < 0.05).

**Table 3 polymers-14-01096-t003:** Neutral and amino sugar contents (molar ratio) of purified fractions.

Sugar Components	Fractions		
	F1	F2	F3
Fucose	ND	0.19	ND
Rhamnose	ND	0.28	ND
Arabinose	ND	ND	ND
Galactose	ND	3.19	1.60
Glucose	1.00	1.00	1.00
Mannose	ND	0.72	0.87
Amino sugars			
N-acetyl galactosamine	0.01	2.23	1.48
N-acetyl glucosamine	0.01	4.16	6.56

ND, not detected.

**Table 4 polymers-14-01096-t004:** Disaccharide composition of the ascidian tunic purified GAGs fractions ^h^.

Peak No. ^a^	Disaccharide	t_R_(min) ^b^	Proportion of the Disaccharides ^c^			
			F2_abc_ ^d^	F2_ac_ ^e^	F3_abc_ ^f^	F3_ac_ ^g^
1	Di-OS	27.35	ND	ND	ND	ND
2	Di-6S	47.79	47.26	49.32	49.32	43.10
3	Di-4S	51.40	52.74	50.68	50.68	56.90
4	Di-SD	74.93	ND	ND	ND	ND
5	Di-SE	77.95	ND	ND	ND	ND
6	Di-SB	78.49	ND	ND	ND	ND
7	Di-TriS	85.66	ND	ND	ND	ND

^a^ Standard peak number in order of elution. ^b^ Retention time of each peak. ^c^ Areas under the peaks were integrated to obtain the disaccharide composition. ^d^ Ascidian fraction 2 treated with chondroitinase ABC. ^e^ Ascidian fraction 2 treated with chondroitinase AC. ^f^ Ascidian fraction 3 treated with chondroitinase ABC. ^g^ Ascidian fraction 3 treated with chondroitinase AC. ^h^ Average peak area of fraction 2: peak 2 (Di-6S) is 48.29 [(47.26 + 49.32)/2]; peak 3 (Di-4S) is 51.71 [(52.74 + 50.68)/2]; Average peak area of fraction 3: peak 2 (Di-6S) is 46.21 [(49.32 + 43.10)/2]; peak 3 (Di-4S) is 53.79 [(50.68 + 56.90)/2] in the HPLC chromatogram. The approximate ratio between Di-6S and Di-4S is 48:52 for fraction 2 and 46:54 for fraction 3. ND, not detected.

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
