# Peer review of "Hair Growth-Promoting Activities of Glycosaminoglycans Extracted from the Tunics of Ascidian (Halocynthia roretzi)"

_polymers, 2022, doi:10.3390/polym14061096_

Round 1
Reviewer 1 Report
This manuscript demonstrated the extraction and purification of Glycosaminoglycans from ascidian tunics and examined its hair growth promoting activity. Overall, the work is well designed, and the conclusion is supported by the findings. I would recommend this work can be published in Polymers if the authors could address my concerns below.
1) Line 16-17: What do “F1”, “F2”, “F3”, “Di-6S”, “Di-6S” mean? The detailed specification for the abbreviation should be provided in the Abstract.
2) The originality of the paper needs to be further clarified in the Introduction part. Some sentences must be added to highlight the importance of investigation to justify the novelty of a high-quality journal paper.
3) The texts in the figures should be modified with a better resolution.
4) The authors are suggested to improve the Conclusion by incorporating some quantitative data and a clear summary.
5) Language should be thoroughly revised as some of the sentences are confusing and some errors can be found.
Author Response
1) Line 16-17: What do “F1”, “F2”, “F3”, “Di-6S”, “Di-6S” mean? The detailed specification for the abbreviation should be provided in the Abstract.
Response 1: Thank you for pointing this concern. The following terms were defined, accordingly. Kindly see in the abstract of revised manuscript.
2) The originality of the paper needs to be further clarified in the Introduction part. Some sentences must be added to highlight the importance of investigation to justify the novelty of a high-quality journal paper.
Response 2: Thank you for the suggestion. Introduction part was modified consequently. Kindly see the revised manuscript.
3) The texts in the figures should be modified with a better resolution.
Response 3: Thank you for pointing this out. Figures were changed higher resolution and clearer text labels. We would also like to clarify that the chromatograms for fractions, F1 and F2, got interchanged along the process. We already changed it appropriately in case it caused confusion to the editors and reviewers. Kindly check in the revised manuscript.
4) The authors are suggested to improve the Conclusion by incorporating some quantitative data and a clear summary.
Response 4: Thank you for the advice. Conclusion was revised accordingly. Kindly check in the revised manuscript.
5) Language should be thoroughly revised as some of the sentences are confusing and some errors can be found.
Response 5: Thank you for the comment. We have carefully checked and revised the whole manuscript as per reviewer’s suggestion. Kindly check the revised manuscript.

Reviewer 2 Report
In this work, the ascidian FFM (F2 and F3 mixture) fraction was demonstrated to delay hair 367 baldness by inhibiting DHT production. In addition, hair re-growth, as well as skin dark-368 ness, was increased in mice treated with FFM. However, the following issues should be addressed.
1) How to ensure the biological activity of the components extracted from ascidian tunics under high temperature (121°C) or by enzymes.
2) Human follicle dermal papilla (HFDP) cells were used to determine the effect of each fraction of the ascidian tunic on cell viability in Figure 2. Why did you choose hyaluronic acid as the control? In addition, 24 hours is too short to reflect the tunic fractions' effect on cells.
3) Text inside figure 1 is unreadable even when magnified. The text size in panels should be increased.
4) The authors claim that “Histologic examination revealed that hair follicles reached the subcutaneous muscle layer, and the hair shaft surrounded by the hair canal emerged through the epidermis in the Minoxidil or FFM treated group.” However, There were no histological results.
5) Why do some groups have six mice and some have only five in Figure 5?
Author Response
1) How to ensure the biological activity of the components extracted from ascidian tunics under high temperature (121°C) or by enzymes.
Response 1: Thank you for this question. Evaluation of antioxidant activities and other biocompatibility assays can be performed to investigate and ensure the biological activity of components extracted from ascidian tunics under high temperature or by enzymes. In addition, inclusion of impurities (which can interfere bioactivities) during the extraction process should be minimized as much as possible. However, this can be tackled in another study which focuses on the optimization of the extraction method of bioactive components from ascidian tunics. In this study, we focused on extraction of glycosaminoglycans (GAGs) from ascidian tunics. GAGs are versatile compounds and can withstand high temperature extraction methods. There have been published studies that used extraction at high temperature or enzymes to extract GAGs from other marine sources wherein GAGs’ biological activities were retained. We believe this also applies to the crude GAGs extracted from ascidian tunics used in this study. Otherwise, the purified fractions wouldn’t exhibit improved cell proliferation on hair follicle papilla cells and demonstrate hair growth promoting activity. We hope this answers the reviewer’s query.
2) Human follicle dermal papilla (HFDP) cells were used to determine the effect of each fraction of the ascidian tunic on cell viability in Figure 2. Why did you choose hyaluronic acid as the control? In addition, 24 hours is too short to reflect the tunic fractions' effect on cells.
Response 2: Thank you for this question. We used hyaluronic acid (HA) as the positive control because HA has the most basic structure among GAGs and functions as a key component in cell proliferation and migration. Additionally, the cell viability assay was done while we were also still analyzing the identity and structure of each fraction. And at that time, we were still confirming whether each fractions contain sulfated glycosaminoglycans (GAGs) or not. We were also considering the potential use of ascidian tunic GAGs in topical applications so believe that hyaluronic acid was appropriate to be used as positive control. We hope this answers the question of the reviewer. Again, thank you for this query.
3) Text inside figure 1 is unreadable even when magnified. The text size in panels should be increased.
Response 3: Thank you for pointing this out. Figures were changed higher resolution and clearer text labels. We would also like to clarify that the chromatograms for fractions, F1 and F2, got interchanged along the process. We already changed it appropriately in case it caused confusion to the editors and reviewers. Kindly check in the revised manuscript.
4) The authors claim that “Histologic examination revealed that hair follicles reached the subcutaneous muscle layer, and the hair shaft surrounded by the hair canal emerged through the epidermis in the Minoxidil or FFM treated group.” However, There were no histological results.
Response 4: Thank you for mentioning this. Histological examination was done just like in normal tissue observation. However, there were no significant differences among the samples that were clearly visible. This resulted to all the samples eventually discarded. Thus, we could not provide the data now. All the authors decided to remove the said phrase/paragraph as it could bring confusion to readers, like how the reviewer pointed it out. We hope that this answered the reviewer’s query.
5) Why do some groups have six mice and some have only five in Figure 5?
Response 5: Thank you for this question. In the duration of the animal experiment, some mice from the treated group (specifically, the group treated with Minoxidil) developed allergy on the 3rd week. That is the reason why some of the pictures only showed 5 mice while other groups still have 6 mice at the end of the experiment. We have indicated it in the figure description. Kindy check in the revised manuscript.